# Spatiotemporal Spiking Entropic Bottleneck: Data-efficient Learning with Joint Redundancy Reduction in Spiking Neural Networks

## Abstract

Spiking neural networks (SNNs) are energy-efficient brain-inspired models, which have received increasing attention in recent years. However, existing SNNs tend to overlook more challenging scenarios with insufficient sample sizes. In data-scarce scenarios, the spatiotemporal dynamics in SNNs often involve joint spatiotemporal redundancy, which results in compromised generalization and reduced robustness. The information bottleneck principle has demonstrated powerful spatial compression in artificial neural networks, but its direct application to SNNs is nontrivial: the discrete, timing-dependent nature of spikes makes spatiotemporal entropy estimation inherently challenging. To reduce the joint redundancy for data-efficient learning, we propose the spatiotemporal spiking entropic bottleneck (STSEB) framework that jointly compresses spatial and temporal information while preserving task-relevant features. Central to STSEB is the spike time matrix, which records each neuron's first spiking time to extract the most critical temporal feature, discard redundant spikes, and align activities across neurons. We further develop a spike-time-matrix-based Rényi's $\alpha$-entropy estimator that captures the intrinsic frequency distribution of spatiotemporal spiking patterns to drive compression under spatiotemporal bottleneck objective. We prove that STSEB obtains more compact latent representations than traditional information bottleneck by average spiking rate and total correlation metrics. The experimental results show that STSEB achieves superior generalization and robustness compared to SOTA under scarce samples, with higher sample efficiency and reduced power consumption. The code will be released upon acceptance.

## 1 Introduction

Spiking neural networks (SNNs) represent an intriguing class of brain-inspired computational models, emulating the intricate communication mechanisms of biological neurons through discrete and sparse spikes Yao et al. (2023b); Li et al. (2021). Their capability for low-power operation and natural compatibility with neuromorphic hardware has increasingly captured the attention of researchers, highlighting their significant potential in contemporary artificial general intelligence Fang et al. (2021); Yin et al. (2021). However, despite ongoing theoretical and hardware advancements in SNNs, challenges remain in their generalization ability and learning efficiency, especially in data-scarce scenarios. Unlike conventional artificial neural networks (ANNs), the discrete spikes in SNNs inherently exhibit sparsity and dynamic complexity. While this allows for efficient, event-driven computation Li et al. (2021); Shen et al. (2025b); Wei et al. (2023), it introduces spatiotemporal joint redundancy that degrades the generalization and robustness of SNNs, particularly in data-scarce scenarios. From the perspective of spatial dimension, neighboring neurons are often connected to adjacent or overlapping input regions, resulting in multiple neurons to encode similar features or patterns and produce highly similar responses Krunglevicius (2015); Saunders et al. (2019); Vertes & Duke (2010); Zhou et al. (2024). It induces spatial redundancy resulting from the spatial topological correlation and overlapping local receptive fields. From the temporal perspective, certain input features may persist over time or change slowly, causing neurons to repeatedly spike over consecutive time steps. This induces temporal redundancy resulting from repeated activations in response to sustained input Yao et al. (2023a); Ponghiran & Roy (2022); Liu et al. (2022); Comşa et al. (2021);

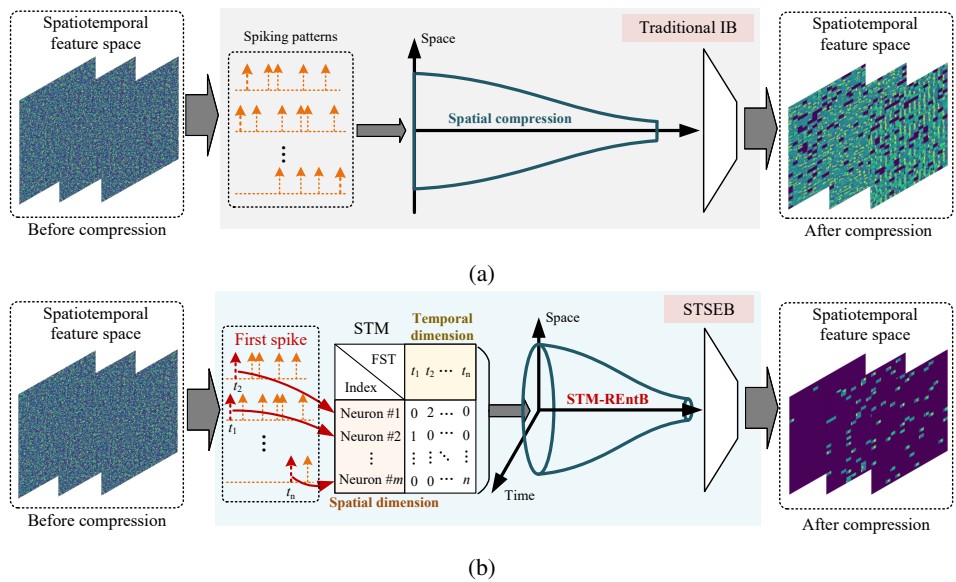

Figure 1: Overall working mechanism of the STSEB method. (a) Traditional IB method with difficulty in spatiotemporal entropy estimation. (b) STSEB realizes the spatiotemporal compression by STM-based Rényi's $\alpha$-entropy functional estimator, which can achieve a more compact latent representation.

Vicente-Sola et al. (2025); Kugele et al. (2020). Effectively compressing this spatiotemporal redundancy is crucial for SNNs to improve the generalization and robustness and enable data-efficient learning.

Information compression seeks to preserve crucial information by eliminating redundant or irrelevant data, thus enhancing generalization and robustness Lee et al. (2021); Shwartz Ziv & LeCun (2024); Hu et al. (2025); Li et al. (2023b). In recent years, the information bottleneck (IB) principle has garnered significant attention due to its success in providing an elegant theoretical framework for achieving optimal information compression in deep neural networks Kawaguchi et al. (2023); Pan et al. (2021). In traditional ANNs that are characterized by continuous and analog signals, this principle has demonstrated significant advantages Li et al. (2022; 2023a). There have been several studies on applying IB in SNNs in various types of training schemes, such as ANN-to-SNN conversion Zhang et al. (2022), supervised direct training by surrogate gradient learning Yang & Chen (2023a;b), and self-supervised training for optical flow estimation Yang et al. (2024b). It is a pity that these studies typically use conventional mutual information (MI) estimation methods originally designed for spatial information compression as illustrated in Figure 1a, which are inadequate for handling the discrete and timing-dependent spikes.

These challenges of extending the IB principle to SNNs arise from the intrinsic sparseness, discreteness, and timing-dependency of SNNs. First, spikes are discrete binary events that are inherently sparse and unevenly distributed Rathi & Roy (2024); Ghosh-Dastidar & Adeli (2009); Yao et al. (2024), which conflicts with the continuous Gaussian distribution assumption that traditional IB rely on Oh et al. (2018); Chang et al. (2020). Second, spike generation in SNNs depends not only on which neuron fires but also on when it fires, introducing spatiotemporal dependency Taherkhani et al. (2020); Eshraghian et al. (2023); Meng et al. (2022). The resulting spatiotemporal joint distributions are highly complex, and directly computing the entropy of spatiotemporal joint distributions requires addressing an extremely high-dimensional state space. It makes computation on the spatiotemporal joint distribution intractable in practical applications, thus making it challenging for traditional IB to compress the spatiotemporal joint entropy. Consequently, this results in suboptimal compression effects with IB, potentially distorting critical spiking patterns, with unresolved spatiotemporal joint redundancy further hindering the efficiency and compactness of information representation within SNNs.

To tackle the problems, we propose a novel training method, called spatiotemporal spiking entropic bottleneck (STSEB), with the key objective of jointly compressing spatial and temporal information while preserving task-relevant features. To this end, the STSEB method first encodes neuronal spike trains into a spike time matrix (STM), in which only the first spike time (FST) of each neuron is recorded, as illustrated in Figure 1b. This provides a natural timestamp that captures the most salient temporal features while eliminating redundant information caused by repeated spiking, and enables temporal alignment across spiking neurons. As noted by Thorpe et al. (2001), early spikes tend to carry the most discriminative information. Timing of the first spike has been shown to be highly indicative of neuronal sensitivity to input stimuli Thorpe et al. (2001); Han & Roy (2020); Yang et al. (2023a). Therefore, it establishes a sparse and unified encoding scheme with the dimensionality of the state space shrunk.

To effectively extract spatiotemporal information from the STM, we design a specialized Rényi's $\alpha$-entropy estimator for spatiotemporal entropy estimation. In this estimator, Rényi's $\alpha$-entropy is employed to deal with the reliance on continuous density estimation by traditional entropy estimation methods, allowing for direct computation of entropy robustly based on the frequency distribution of the sparse and irregular spiking patterns. Building upon the proposed estimator, we derive a novel information bottleneck objective from a first principle, termed STM-REntB, yielding spatiotemporal information compression with spatiotemporal joint redundancy reduction. With average spiking rate and total correlation evaluations, we prove that STSEB enables the latent representations more compact compared to traditional IB. Based on STSEB, we outperform existing SOTA SNN direct training methods, as well as the latest information bottleneck approach HOSIB Yang & Chen (2023a), on the CIFAR-10, CIFAR-100, and DVS-Gesture datasets under varying levels of training sample scarcity, demonstrating the generalization capability and sample efficiency of STSEB in data-scarce scenarios. Additionally, after introducing Gaussian noise, as well as black-box and white-box adversarial attacks to DVS-Gesture, STSEB demonstrates stronger robustness than SOTA methods across training sets of different scales. Compared to traditional IB, STSEB also reduced power consumption by 5.39%.

## 2 RELATED WORK

The problem of information compression represents a critical research focus within the field of deep neural networks Cheng et al. (2018). Information-theoretic learning can provide fundamental solutions to information compression challenges Hild et al. (2006); Deng et al. (2016), with the IB theory standing out as one of its most advanced and representative frameworks. The core principle of the IB theory is that during the representation learning phase, a network should compress the input data as much as possible while retaining sufficient information to support the target task, thereby enhancing the generalization and efficiency Kawaguchi et al. (2023); Hu et al. (2024). There are three primary approaches to applying the IB in SNNs.

First, IB is integrated with principles from neuroscience, utilizing these principles to derive learning rules based on neuroscience mechanisms research. IB has been proposed using stochastic spiking neurons with refractory periods Klampfl et al. (2009); Buesing & Maass (2010). By utilizing IB strategy, three-factor learning rule is proposed with a local Hebbian component and a global modulatory signal Klampfl et al. (2006); Daruwalla & Lipasti (2024). Moreover, SpiKL-IP applies information-theoretic approach to intrinsic plasticity by maximizing the entropy of the firing rate distribution toward a target optimal exponential distribution Zhang & Li (2019).

Second, IB is incorporated into ANN-to-SNN conversion training strategies. The core idea is to first train a traditional ANNs using an IB loss, and then convert the trained ANN into an SNN model with structurally and weight-wise equivalent properties. Following this approach, 2O-IB optimizes the latent representations within the ANN through IB-based training before converting the network into an SNN Zhang et al. (2022). In theory, this strategy can leverage a wide range of IB techniques that have been extensively validated in deep learning Ma et al. (2020); Nguyen & Choi (2019); Ngampruetikorn & Schwab (2022).

Third, there are also some works that apply IB to direct training strategies for SNNs based on surrogate gradient learning. Building on this concept, a series of IB methods and their high-order variants have been proposed, including SIBoLS Yang et al. (2023b), SNIB Yang & Chen (2023b), HOSIB Yang & Chen (2023a), and HHO-IB Wu et al. (2025). Although IB is not explicitly used, IM-Loss

introduces an information maximization loss function to address the issues of spike information loss and accuracy degradation Guo et al. (2022). SMEIL employs the maximum entropy principle to promote perturbation of the underlying source distribution, thereby increasing the predictive uncertainty of the current model Yang et al. (2024a).

Although these methods provide valuable insights, we believe further improvements are necessary. Firstly, the aforementioned approaches typically use the assumption of traditional IB distributions, which are limited in their application to the sparse spatiotemporal information flow in SNNs. Secondly, these methods have not explicitly measured the high-dimension spatiotemporal information and consequently calculated the information entropy, making it challenging to further compress the redundancy in the spatiotemporal information dimension. In this paper, we will carefully consider these issues and propose effective solutions.

## 3 PRELIMINARY

### 3.1 SPIKING NEURAL NETWORKS

SNNs are computational models inspired by biological neural networks, where neurons communicate through discrete spikes rather than continuous signals. The fundamental unit of SNNs is the spiking neuron, which generates spikes when the membrane potential surpasses a certain threshold. This spiking behavior is captured by models like the leaky integrate-and-fire (LIF) neuron model, which is widely used in SNNs to simulate the dynamics of real neurons.

In the LIF model, the membrane potential $V[t]$ is computed by summing the previous state $H[t-1]$ and the input current $I[t]$. The neuron fires a spike if $V[t]$ exceeds a threshold $V_{th}$, with the spiking behavior represented by a binary function $J[t] = \text{Heaviside}(V[t] - V_{th})$. If the neuron spikes, its membrane potential is reset to a resting value $V_{\text{reset}}$. Otherwise, it maintains the current potential. The updated membrane potential $H[t]$ is thus computed as:

$$H[t] = V_{\text{reset}} \cdot J[t] + V[t] \cdot (1 - J[t]). \tag{1}$$

### 3.2 INFORMATION BOTTLENECK

The IB theory is a fundamental concept in information theory applied to deep learning. Its core idea is to compress the input while preserving the most relevant features for predicting the output. Given an input variable $X$ and output variable $Y$, the theory constructs a Markov chain $X \rightarrow Z \rightarrow Y$, where the intermediate variable $Z$ represents a compressed representation of $X$. The mapping from $X$ to $Z$ is denoted by the conditional distribution $P(Z|X)$, which induces a marginal distribution over $Z$: $Z \sim P(Z) = \int P_{Z|X}(z|x) p_X(x) \, dx$. The goal of IB is to retain as much information about $Y$ as possible in $Z$ while minimizing the information flow from $X$ to $Z$. This trade-off can be formulated as the following optimization problem:

$$\max_{Z \in \mathcal{A}} I(Z; Y) \quad \text{s.t.} \quad I(X; Z) \leq \epsilon, \tag{2}$$

where $I(\cdot; \cdot)$ denotes MI, and $\mathcal{A}$ is the set of all possible mappings $Z \sim P(Z|X)$. Introducing a Lagrange multiplier $\beta$ yields the IB Lagrangian objective as follows:

$$L_{\text{IB}} = \max_{Z \in \mathcal{A}} I(Z; Y) - \beta I(X; Z), \tag{3}$$

where $\beta$ controls the trade-off between compression and prediction. A larger $\beta$ enforces a stronger compression constraint. To compute MI, the Kullback–Leibler (KL) divergence is widely used, which measures the similarity between two distributions. specifically, MI can be approximated as follows:

$$I_{P(X),P(Z|X)}(X; Z) = \sum_{x,z} P_{Z|X}(z|x) p_X(x) \log \frac{P_{Z|X}(z|x)}{P_Z(z)}$$

$$\approx \frac{1}{N} \sum_{n=1}^{N} D_{\text{KL}}(P_{Z|X}(z|x_n) \| R(Z)), \tag{4}$$

where $R(Z)$ is a variational approximation to the marginal distribution $P(Z)$, and $N$ is the number of training samples.

## 4 THE STSEB FRAMEWORK

In IB theory, the objective of the model is to minimize $I(X; Z)$, thereby compressing the redundant information in the input $X$ while retaining the most crucial features. However, for SNNs with spatiotemporal dimensions, the traditional IB that only compresses $I(X; Z)$ fails to capture the information entropy along the temporal dimension, thus limiting its ability to effectively compress spatiotemporal joint redundancy. Moreover, the assumption of continuous signal distributions in IB is evidently unsuitable for SNNs, where discrete distribution assumptions more accurately capture the feature information. To address these challenges, we define a Rényi's $\alpha$-entropy estimator based on the STM to characterize the spatiotemporal information of SNNs, and introduce the STSEM framework, which aligns with the spatiotemporal discrete features of SNNs. Then, we present the generalization and sample complexity bounds of STSEB and define metrics for spatiotemporal information compression and redundancy. We also provide a visualized comparison with the traditional IB method.

### 4.1 STM BASED RÉNYI'S $\alpha$-ENTROPY ESTIMATOR AND STSEB OBJECTIVE

In the STSEB framework, to achieve compression of redundant information along the spatiotemporal dimension, we first describe the spatiotemporal information. Spike trains in SNNs are inherently temporal, with the temporal dimension capturing the precise firing times of each neuron. Compared to the average firing rate, first spikes provide a sparser representation with minimal information redundancy, preserving the temporal structure. We define the STM $T_Z$ based on the first spiking time of each neuron as follows:

$$T_Z = \begin{bmatrix} T_{11} & \cdots & T_{1T} \\ \vdots & \ddots & \vdots \\ T_{N1} & \cdots & T_{NT} \end{bmatrix} \in \mathbb{R}^{N \times T}, \tag{5}$$

$$T_{nt} = \begin{cases} \min \{t \in [1, T] \mid O_{nt} = 1\} & \text{if such } t \text{ exists,} \\ 0 & \text{else,} \end{cases} \tag{6}$$

where $O_{nt}$ represents the output of the $n$-th neuron at the $t$-th time step in the bottleneck layer. $T_Z$ records the FST for each neuron, capturing fine-grained temporal information. After obtaining the STM, to achieve compression of spatiotemporal redundant information, it is necessary to measure the spatiotemporal information content of the intermediate variable $Z$. In information theory, Shannon entropy is commonly used to quantify the amount of information contained in a variable. Rényi's $\alpha$-entropy, on the other hand, is a generalization of Shannon entropy, offering improved numerical stability and extensibility, particularly for high-dimensional information. Based on the STM, we construct an estimator using Rényi's $\alpha$-entropy to represent the spatiotemporal information content of intermediate variable $Z$.

**Definition 1 (STM based Rényi's $\alpha$-Entropy Estimator)** *Given a kernel function $k : \mathcal{X} \times \mathcal{X} \to \mathbb{R}$ and an unlimited number of kernels, the Gram matrix $K$ for STM $T_Z$ can be calculated as $K_{ij} = k(T_{Zi}, T_{Zj})$. The normalised positive semi-definite matrix $A$ can then be computed as $A_{ij} = \frac{K_{ij}}{\sqrt{K_{ii}K_{jj}}}$. The STM based Rényi-$\alpha$ entropy estimator $H_\alpha(T_Z)$ is given by:*

$$H_\alpha(T_Z) = \frac{1}{1-\alpha} \log_2 \left( \text{tr}(A^\alpha) \right)$$

$$= \frac{1}{1-\alpha} \log_2 \left( \sum_{i=1}^{N} \lambda_i(A)^\alpha \right), \tag{7}$$

*where $\lambda_i(A)$ denotes the $i$-th eigenvalue of matrix $A$.*

The STM-based Rényi-$\alpha$ entropy estimator characterizes the spatiotemporal information contained in the intermediate variable $Z$, addressing the gap in IB theory where spatiotemporal information is not considered in SNNs. A smaller value of $H_\alpha(T_Z)$ indicates that the spatiotemporal features of

the intermediate variable $Z$ are more compact. To achieve information compression along the spatiotemporal dimension, we propose the STSEB based on the STM-based Rényi-$\alpha$ entropy estimator, and define its objective, termed STM-REntB, as presented in Definition 2:

**Definition 2 (STSEB Objective: STM-REntB)** *Given the input $X$ and output $Y$ of SNN, an intermediate variable $Z$ is constructed, where the MI between $X$ and $Z$ is $I(X; Z)$, and the MI between $Z$ and $Y$ is $I(Z; Y)$. Based on Definition 1, the Rényi-$\alpha$ entropy of $Z$ can be computed, denoted as $H_\alpha(T_Z)$. The objective of STSEB aims to compress redundant information as much as possible in both spatial and spatiotemporal dimensions to obtain effective and compact spatiotemporal features. The mathematical form of the objective is as follows:*

$$\mathcal{L}_{STM\text{-}REntB} = \max_{Z \in \Delta} I(Z; Y) - \beta I(X; Z) - \gamma H_\alpha(T_Z), \tag{8}$$

*where $\beta$ and $\gamma$ respectively control the extent of compression in the spatial and spatiotemporal domains.*

In the STSEB objective, $\beta$ and $\gamma$ serve as trade-off parameters, where larger values correspond to stronger compression along the respective dimension. By adjusting these hyperparameters, STSEB achieves a balance between spatial and temporal feature abstraction, resulting in more compact yet informative representations, and enhancing the generalization and robustness of SNNs. The pseudocode for STSEB is provided in the Appendix A.5.

## 4.2 THEORETICAL GUARANTEES AND METRICS OF STSEB

To demonstrate the effectiveness of STSEB, we theoretically derive its objective's generalization and sample complexity bounds. Consider an SNN model with input-output pairs denoted as $x$ and $y$, where each sample is independently drawn from an unknown distribution $\mathcal{D}$. Let the model family be $\mathcal{F} = \{Z_\theta : \theta \in \Theta\}$, where $Z_\theta : \mathcal{X} \to \mathcal{Z}$ is a mapping from model inputs to latent variables. Let $\ell(Z; x, y)$ denote the loss function. The expected risk is defined as: $L(Z) = \mathbb{E}_{(x,y) \sim \mathcal{D}} [\ell(Z; x, y)]$, and the empirical risk is: $\hat{L}_n(Z) = \frac{1}{n} \sum_{i=1}^n \ell(Z; x_i, y_i)$. Define the composite loss class as: $\mathcal{L} \circ \mathcal{F} = \{(x, y) \mapsto \ell(Z; x, y) : Z \in \mathcal{F}\}$. Then the Rademacher complexity of this class is given by:

$$\mathcal{R}_n(\mathcal{L} \circ \mathcal{F}) = \mathbb{E}_{\sigma,(x_i,y_i)} \left[ \sup_{Z \in \mathcal{F}} \frac{1}{n} \sum_{i=1}^n \sigma_i \ell(Z; x_i, y_i) \right], \tag{9}$$

where $\sigma_i$ are independent Rademacher random variables. We make the following assumption:

**Assumption 4.1.** There exists a constant $B > 0$ such that for all $(x, y)$ and $Z \in \mathcal{F}$, the loss function satisfies:

$$0 \leq \ell(Z; x, y) \leq B. \tag{10}$$

In practice, loss functions are bounded and usually decrease during training. Hence, this assumption is reasonable and commonly satisfied. Based on the above definitions and Assumption 1, we now present the generalization and sample complexity bounds:

**Theorem 1 (Generalization Bound)** *For any $\delta > 0$, with probability at least $1 - \delta$, the following holds for all $Z \in \mathcal{F}$:*

$$L(Z) \leq \hat{L}_n(Z) + 2\mathcal{R}_n(\mathcal{L} \circ \mathcal{F}) + B\sqrt{\frac{\ln(1/\delta)}{2n}}. \tag{11}$$

**Theorem 2 (Sample Complexity Bound)** *Let $Z^* = \arg\max_Z L(Z)$ and $\hat{Z} = \arg\max_Z \hat{L}_n(Z)$. If the number of samples $n$ satisfies:*

$$n \geq \frac{C}{\epsilon^2} \left( \mathcal{R}_n(\mathcal{L} \circ \mathcal{F})^2 + \ln\frac{1}{\delta} \right), \tag{12}$$

*for some constant $C$, then with probability at least $1 - \delta$, we have: $L(Z^*) - L(\hat{Z}) \leq \epsilon$.*

Theorems 1 and 2 establish the generalization and sample complexity bounds of the STSEB training algorithm. From Eq. 11, we observe that the gap between the training error and generalization

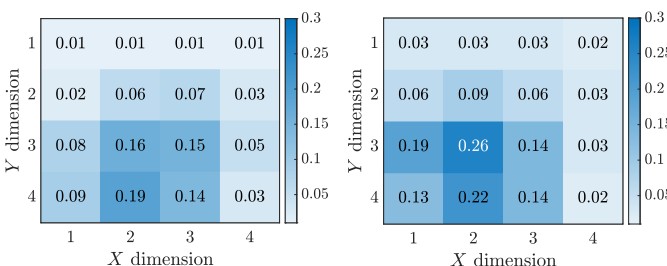 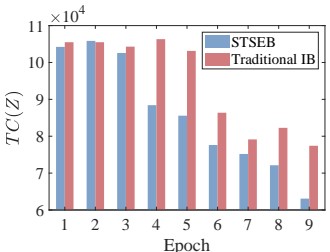

Figure 2: Comparison of FR with STSEB (left) and traditional IB (right). STSEB demonstrates a consistently lower FR, evidencing its superior spatiotemporal joint compression capability compared to traditional IB.

Figure 3: Evolution of $TC(Z)$ during training. STSEB exhibits a lower $TC(Z)$, suggesting its less redundant information.

error of STSEB is controlled by the Rademacher complexity and the sample size $n$. A lower model complexity and larger number of samples lead to a smaller generalization gap, indicating better generalization performance. Under the same sample size, STSEB compresses information across both spatial and temporal dimensions, resulting in more compact representations compared to traditional IB methods. Consequently, it achieves lower model complexity and exhibits improved generalization ability. Furthermore, Eq. 12 indicates that STSEB can achieve near-optimal performance as long as the sample size meets a sufficient threshold. Proofs of the above theorems are provided in the Appendix A.6.

To quantify and more intuitively assess the spatiotemporal information compression and redundancy of the latent variable $Z$ in STSEB, and to compare it with the traditional IB method, we propose corresponding metrics and conduct analysis. For spatiotemporal information compression, we use the firing rate (FR) of the latent variables as the metric. For redundancy, we use $TC(Z)$ to measure the redundancy of information contained in the latent variables. The specific definitions can be found in the Appendix A.7. Figure 2 shows the FR heatmaps for the intermediate layer of STSEB and traditional IB. It can be observed that STSEB exhibits a lower FR, indicating higher spatiotemporal compression. Figure 3 demonstrates the changes in $TC(Z)$ for STSEB and traditional IB during training. It shows that $TC(Z)$ decreases over training and consistently remains lower than that of IB, proving that the latent variables in STSEB, after spatiotemporal compression, contain less redundant information.

## 5 EXPERIMENT

In this section, we perform classification tasks on both static and neuromorphic datasets and conduct experiments with different training set sizes to validate the generalization and robustness of the proposed STSEB, especially in data-scarce scenarios. We compare STSEB with other methods applied to optimize the SNN training process and also introduce Gaussian noise, black-box, and white-box adversarial attacks to evaluate the model's robustness under data-sparse conditions. Finally, we benchmark the energy consumption performance of STSEB, comparing it with the baseline SNN model and traditional IB methods. Detailed experimental configurations are provided in the Appendix A.4.

### 5.1 PERFORMANCE OF STSEB ON STATIC AND NEUROMORPHIC DATASETS

We test STSEB on the DVS-Gesture, CIFAR-10, and CIFAR-100 datasets with different training set sizes and compare it with other training optimization methodsZhu et al. (2024); Liang et al. (2025); Yang & Chen (2023a); Duan et al. (2022), including a comparison with HOSIBYang & Chen (2023a) to demonstrate the effectiveness of STSEB's spatiotemporal joint compression. The experimental results, as shown in Figure 4, indicate that our method achieves optimal results, especially in scenarios with scarce training data, highlighting the data-efficiency of STSEB. Additionally, to further investigate the advantages of STSEB in compressing spatiotemporal redundant information, we demonstrate the feature maps of the intermediate variables in STSEB across multiple time steps.

As shown in Figure 5, in the feature maps corresponding to the intermediate variables in STSEB, the features at the same time step are sparser compared to traditional IB methods, indicating that STSEB compresses more redundant information in the spatial dimension. For the same location at different time steps (T=1 to T=4), STSEB exhibits significantly fewer repeated spikes at adjacent time steps compared to the traditional IB method. This indicates that STSEB is able to compress more temporal redundant information caused by repeated spikes from neurons, confirming the effectiveness of the STM-based Rényi's $\alpha$-Entropy Estimator in compressing spatiotemporal joint redundancy.

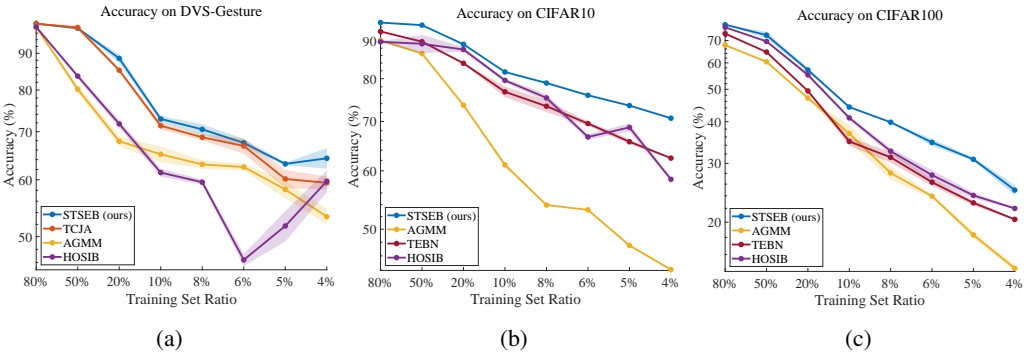

Figure 4: Comparison with the SOTA methods on DVS-Gesture, CIFAR-10, and CIFAR-100 datasets under varying training set ratios.

## 5.2 ROBUSTNESS ANALYSIS OF STSEB

To investigate the improvement in SNN robustness through spatiotemporal joint redundancy compression by STSEB, we conduct experiments on the DVS-Gesture dataset under various noise conditions, including Gaussian noise, black-box, and white-box adversarial attack noise. Detailed information about the noise deployment is provided in the Appendix A.8. We compared our method with TCJA Zhu et al. (2024), which performs second best on the DVS-Gesture dataset as shown in Figure 4. The experimental results under three types of noise are shown in the Table 1. Under all noise conditions, STSEB consistently achieves higher accuracy, especially under white-box adversarial attack noise, where the average accuracy improvement (Avg. Imp.) over TCJA reaches 23.73% at the sample ratio of 0.1. The robustness experiments demonstrate that STSEB, by compressing spatiotemporal information, enables the model to extract more important and compact features, significantly improving the robustness of SNNs in noisy environments. This improvement is particularly noticeable in data-scarce scenarios.

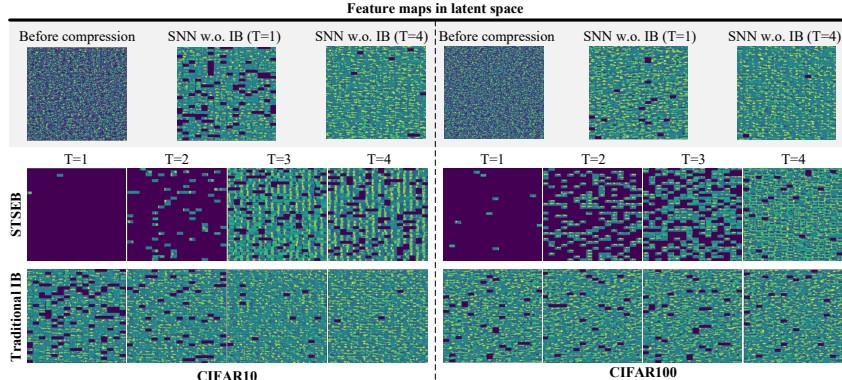

Figure 5: Comparison of feature maps in latent space. STSEB exhibits sparser feature maps compared to traditional IB, indicating the acquisition of more compact spatiotemporal features.

Table 1: Performance comparison with the SOTA method under different noise types and sample ratios.

| Noise type | Method | Sample Ratio | | | | |
|---|---|---|---|---|---|---|
| | | 1 | 0.5 | 0.2 | 0.1 | 0.05 |
| Gaussian | TCJA | $56.43\% \pm 0.042$ | $48.05\% \pm 0.136$ | $34.55\% \pm 0.130$ | $22.74\% \pm 0.027$ | $22.05\% \pm 0.106$ |
| | **STSEB** (ours) | $\mathbf{58.51\%} \pm 0.081$ | $\mathbf{51.22\%} \pm 0.125$ | $\mathbf{50.0\%} \pm 0.059$ | $\mathbf{36.15\%} \pm 0.128$ | $\mathbf{39.62\%} \pm 0.045$ |
| | Avg. Imp. | 2.08% ↑ | 3.17% ↑ | 15.45% ↑ | 13.41% ↑ | 17.57% ↑ |
| Black-box | TCJA | $39.21\% \pm 0.014$ | $30.4\% \pm 0.026$ | $35.51\% \pm 0.019$ | $20.47\% \pm 0.029$ | $13.35\% \pm 0.002$ |
| | **STSEB** (ours) | $\mathbf{51.72\%} \pm 0.0051$ | $\mathbf{30.90\%} \pm 0.005$ | $\mathbf{39.58\%} \pm 0.015$ | $\mathbf{44.2\%} \pm 0.013$ | $\mathbf{29.12\%} \pm 0.006$ |
| | Avg. Imp. | 12.51% ↑ | 0.50% ↑ | 4.07% ↑ | 23.73% ↑ | 15.77% ↑ |
| White-box | TCJA | $80.38\% \pm 0.033$ | $71.59\% \pm 0.001$ | $40.62\% \pm 0.002$ | $32.16\% \pm 0.0003$ | $29.58\% \pm 0.053$ |
| | **STSEB** (ours) | $\mathbf{81.58\%} \pm 0.0243$ | $\mathbf{72.54\%} \pm 0.01$ | $\mathbf{41.89\%} \pm 0.012$ | $\mathbf{34.31\%} \pm 0.0004$ | $\mathbf{36.79\%} \pm 0.0004$ |
| | Avg. Imp. | 1.20% ↑ | 0.95% ↑ | 1.27% ↑ | 2.15% ↑ | 7.21% ↑ |

## 5.3 ABLATION STUDY

We conduct the ablation study on the components of STSEB using the DVS-Gesture dataset with 5% of the training set, testing the impact of each component on the generalization and robustness of the SNN. The experimental results, as shown in Table 2, indicate that the introduction of the STM-based Rényi's $\alpha$-Entropy Estimator improves the model's accuracy, both on the clean CIFAR-10 dataset and the CIFAR-10 dataset with added Gaussian noise. This demonstrates that the STM-based Rényi's $\alpha$-Entropy Estimator effectively enhances the generalization and robustness of the SNN model, improving the model's data efficiency.

Table 2: Ablation study of STSEB on the DVS-Gesture dataset.

| Dataset | $I(X;Z)$ | $H_\alpha(T_Z)$ | ACC |
|---|---|---|---|
| DVS-Gesture with 5% SR | ✗ | ✗ | 60.15% |
| | ✓ | ✗ | 60.83% |
| | ✓ | ✓ | **63.13%** |
| DVS-Gesture with 5% SR under Gaussian noise | ✗ | ✗ | 22.05% |
| | ✓ | ✗ | 24.15% |
| | ✓ | ✓ | **39.62%** |

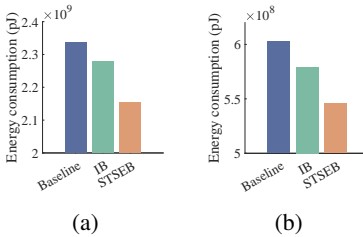

Figure 6: Energy consumption comparison. (a) Comparison on ResNet. (b) Comparison on VGG.

## 5.4 POWER CONSUMPTION ANALYSIS

Building upon prior works Shen et al. (2025a); Shi et al. (2024), we theoretically assess the energy efficiency of STSEB by measuring the number of synaptic operations on the Loihi neuromorphic Davies et al. (2018) . The results, as shown in Figure 6, compare the theoretical energy consumption of the SNN baseline, the traditional IB approach, and STSEB when instantiated on ResNet and VGG backbones. As illustrated, STSEB achieves markedly higher energy efficiency, an outcome attributed to its joint spatio-temporal redundancy compression that yields more compact intermediate representations and concomitantly lowers the spike firing rates across individual neuronal layers. Detailed derivations of the energy model and numerical data are provided in the Appendix A.9.

## 6 CONCLUSION

We propose a training paradigm for SNNs called STSEB, which performs joint redundancy suppression along both spatial and temporal dimensions by constructing a STM–based spatiotemporal matrix and minimizing its Rényi $\alpha$-entropy, thereby yielding maximally compressed yet informative representations. Extensive empirical evaluations reveal that STSEB significantly improves SNN generalization and robustness, enhances data-efficiency in data-scarce scenarios, and concomitantly reduces energy consumption of SNNs.

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

# A APPENDIX

## A.1 ETHICS STATEMENT

This paper focuses on the study of Spiking Neural Networks (SNNs) with the aim of improving the compression of spatiotemporal redundant information and optimizing the generalization and robustness of SNN models. We ensure that all research adheres to the principles outlined in the ICLR Code of Ethics and does not involve any violations of ethical guidelines.

## A.2 REPRODUCIBILITY STATEMENT

We have made efforts to ensure the reproducibility of the results presented in this paper. In Appendix A2, we provide details of the experimental setup. In Appendix A3, we include the corresponding pseudocode for the methods. Additionally, we provide some of the code in the supplementary materials, and once the paper is publicly published, we will release the full code of this work on GitHub for the community to reproduce.

### A.3 LLM Usage Statement

The present study was conducted without the use of any Large Language Models or LLM-based tools throughout its entire process, including conceptualization, experimental design, data processing, result analysis, and manuscript preparation. All text composition, figure generation, and analytical work were independently performed by the authors, relying solely on conventional academic methodologies and human expertise. The findings presented herein represent the original contributions of the research team, without reliance on generative AI systems.

### A.4 Experimental Details

**Static Datasets.** CIFAR-10 and CIFAR-100 Krizhevsky & Hinton (2009) are widely used standardized static datasets in machine vision and deep learning. Both contain 60,000 32×32 pixel color images, with 50,000 images for training and 10,000 images for testing. CIFAR-10 covers 10 general categories (e.g., airplanes, cars, birds), with each category containing 6,000 images. CIFAR-100 extends the dataset by including 100 categories (e.g., apples, mushrooms, whales), with each category providing only 600 images.

**Neuromorphic Datasets.** DVS-Gesture Amir et al. (2017) is a neuromorphic gesture recognition dataset based on the Dynamic Vision Sensor (DVS), specifically designed for event-driven spatiotemporal pattern recognition and SNNs. This dataset includes 29 different gesture actions recorded under varying lighting conditions and background environments, using a DVS camera to capture asynchronous event stream data. Each sample captures dynamic spatiotemporal features of gestures with microsecond-level time resolution, represented as a quadruple (timestamp, pixel coordinates, event polarity), reflecting local changes in brightness during gesture movement. DVS-Gesture contains 1,342 samples, divided into training and testing sets, emphasizing fine-grained segmentation and classification of continuous gesture actions. The challenges of this dataset lie in handling the high sparsity of event streams, noise interference, and temporal dependencies introduced by varying motion speeds. It is widely used to validate the effectiveness of SNNs and supports performance evaluation on low-power neuromorphic hardware.

**Implementation Details.** All experiments were conducted using the PyTorch framework on RTX 4090 and A6000 GPUs. For the neuromorphic dataset DVS-Gesture, we adopted the same architecture as TCJA-SNN Zhu et al. (2024): 128C3-LIF-MP2-128C3-LIF-MP2-128C3-LIF-MP2-128C3-LIF-MP2-128C3-LIF-MP2-0.5DP-512FC-LIF-0.5DP-100FC-LIF-Voting. The model was optimized using Adam with a learning rate of 0.001 and batch size of 64. We employed 10 timesteps for spiking neurons and trained the network for 1,000 epochs. For static image benchmarks (CIFAR-10/100), we utilized the MS-ResNet18 architecture Kim et al. (2025) with input resolution 48×48. The network was trained using SGD with 0.9 momentum and cosine annealing scheduler Loshchilov & Hutter (2017). We set the batch size to 64, learning rate to 0.1, weight decay to 5e-5, and used 6 timesteps for spiking neuronal dynamics. Parameter configurations are provided in Table 3.

Table 3: Training parameters for different datasets.

| Dataset | Optimizer | Batch size | Timestep | Initial LR | Training epoch |
|---------|-----------|-----------|----------|------------|----------------|
| DVS-Gesture | Adam | 16 | 20 | 0.001 | 1000 |
| CIFAR10 | SGD | 64 | 6 | 0.1 | 250 |
| CIFAR100 | SGD | 64 | 6 | 0.1 | 250 |

### A.5 Pseudocode

STSEB quantitatively estimates spatiotemporal information using first-spike-matrix-based Rényi's $\alpha$-entropy based on the first-spike matrix. During the training of the SNN, it compresses spatiotemporal redundant information, leading to more compact spatiotemporal features. This enhances SNN model's generalization ability and robustness. An overview of this process is described in Algorithm 1.

---

**Algorithm 1** Training STSEB

---

**Input:** Training data $X$, labels $Y$, batch size $N$, number of epochs $T$,
hyperparameters $\beta$, $\alpha$, Rényi order $\alpha_{\text{Rényi}}$
**Output:** Trained model parameters $\theta^*$, compressed representation $Z$

1: **for** epoch $t = 1$ to $T$ **do**
2:     Fetch a mini-batch $\{(x^{(i)}, y^{(i)})\}_{i=1}^N$
3:     $Z \leftarrow \text{SNN\_Encode}(X; \theta)$
4:     **for** $i = 1$ to $N$ **do**
5:         **for** neuron $j = 1$ to $d$ **do**
6:             $T_Z[j] \leftarrow \min\{t \mid Z[j][t] = 1\}$
7:             **if** no spike occurs **then**
8:                 $T_Z[j] \leftarrow 0$
9:             **end if**
10:         **end for**
11:     **end for**
12:     Compute Gram matrix $K$ with kernel: $K_{ij} = k(T_{Zi}, T_{Zj})$
13:     Normalize $A_{ij} = \frac{K_{ij}}{\sqrt{K_{ii}K_{jj}}}$
14:     Compute Rényi entropy: $H_{T_Z} = \frac{1}{1-\alpha_{\text{Rényi}}} \log_2\left(\sum_i \lambda_i^\alpha\right)$
15:     Estimate $I(X; Z)$ using Laplace KDE
16:     Estimate $I(Z; Y)$ via classifier or variational method
17:     Compute loss: $L_{\text{STSEB}} = -I(Z; Y) + \beta I(X; Z) + \alpha H_{T_Z}$
18:     Compute gradients $\nabla_\theta L_{\text{STSEB}}$
19:     Update parameters $\theta$ using optimizer
20: **end for**

---

A.6   PROOF

**Proof of Theorem 1.**   Let $\mathcal{D} = \{(x_1, y_1), (x_2, y_2), \ldots, (x_n, y_n)\}$ be a sample drawn i.i.d. from the distribution $P$ over $\mathcal{X} \times \mathcal{Y}$. For any latent variable $Z \in \mathcal{F}$, define the true risk and the empirical risk as follows:

$$L(Z) = \mathbb{E}_{(x,y)\sim P}[\ell(Z; x, y)], \tag{13}$$

$$\hat{L}_n(Z) = \frac{1}{n} \sum_{i=1}^n \ell(Z; x_i, y_i), \tag{14}$$

where $\ell(Z; x, y)$ denotes the loss function. To derive the generalization bound for STSEB, we need to obtain an upper bound for the difference between the true risk $L(Z)$ and the empirical risk. To do this, we introduce the technique of symmetrization by introducing *Rademacher random variables* $\xi_1, \xi_2, \ldots, \xi_n$, where each $\xi_i$ is independent and $\mathbb{P}(\xi_i = \pm 1) = 1/2$. We then construct the symmetrized empirical risk as follows:

$$\hat{L}_n^*(Z) = \frac{1}{n} \sum_{i=1}^n \xi_i \ell(Z; x_i, y_i). \tag{15}$$

Since $\xi_i$ are independent and symmetrically distributed, we have: $\mathbb{E}[\hat{L}_n^*(Z)] = 0$.

Thus, the expectation of the symmetrized empirical risk is zero, and the randomness is effectively controlled through the introduction of $\xi_i$. Now, for each latent variable $Z$, we obtain the key inequality:

$$\mathbb{E}\left[\sup_{Z \in \mathcal{F}} \left(L(Z) - \hat{L}_n(Z)\right)\right] \leq 2\mathcal{R}_n(L(\mathcal{F})), \tag{16}$$

where $\mathcal{R}(L(\mathcal{F}))$ is the Rademacher complexity of the function class $\mathcal{F}$. To further control the deviation between the true risk and the empirical risk, we apply *Hoeffding's inequality*. Hoeffding's inequality provides a bound on the probability of the deviation between the empirical and true risks for each loss function $\ell(Z; x_i, y_i)$, given that the loss is bounded in the interval $[0, B]$. Specifically, for any latent variable $Z$ and any $\epsilon > 0$, we have:

$$P\left(\hat{L}_n(Z) - L(Z) \geq \epsilon\right) \leq \exp\left(-\frac{2n^2\epsilon^2}{B^2}\right). \tag{17}$$

This inequality controls the deviation between the empirical and true risks, and it shows that as the sample size increases, the empirical risk converges to the true risk with high probability. For every latent variable $Z \in \mathcal{F}$, we want to control the deviation for all latent variable in the class. We use the *union bound* to extend this result to all latent variable in $\mathcal{F}$. Applying the union bound, we obtain:

$$P\left(\sup_{Z \in \mathcal{F}} |L(Z) - \hat{L}_n(Z)| \geq \epsilon\right) \leq \sum_{Z \in \mathcal{F}} P\left(|L(Z) - \hat{L}_n(Z)| \geq \epsilon\right). \tag{18}$$

After applying Hoeffding's inequality for each latent variable $Z$, we get:

$$P\left(|L(Z) - \hat{L}_n(Z)| \geq \epsilon\right) \leq \exp\left(-\frac{2n^2\epsilon^2}{B^2}\right). \tag{19}$$

By combining these results, we finally obtain an upper bound for the generalization error. Combining the symmetrization and union bound, we obtain the following generalization bound for the difference between the true and empirical risks:

$$|L(Z) - \hat{L}_n(Z)| \leq 2\mathcal{R}_n(L(\mathcal{F})) + B \cdot \sqrt{\frac{\ln(1/\delta)}{2n}}, \tag{20}$$

where $\mathcal{R}(L(\mathcal{F}))$ is the contribution from the Rademacher complexity, and $\sqrt{\frac{\ln(1/\delta)}{2n}}$ is the probabilistic bound from Hoeffding's inequality.

**Proof of Theorem 2.**  Assume $Z^* = \arg\max_Z L(Z)$ and $\hat{Z} = \arg\max_Z \hat{L}_n(Z)$, based on the optimality of empirical risk minimization, we know that $\hat{L}_n(\hat{Z}) \geq \hat{L}_n(Z^*)$, therefore, we have

$$L(Z^*) - L(\hat{Z}) = (L(Z^*) - \hat{L}_n(Z^*)) + (\hat{L}_n(Z^*) - \hat{L}_n(\hat{Z})) + (\hat{L}_n(\hat{Z}) - L(\hat{Z})).$$

The middle term $\hat{L}_n(Z^*) - \hat{L}_n(\hat{Z}) \leq 0$, which can be ignored. Based on Theorem 1, by using the error bounds, we have

$$L(Z^*) - \hat{L}_n(Z^*) \leq \Delta_n,$$
$$\hat{L}_n(\hat{Z}) - L(\hat{Z}) \leq \Delta_n, \tag{21}$$

where

$$\Delta_n = 2\mathcal{R}_n + B\sqrt{\frac{\ln(1/\delta)}{2n}}. \tag{22}$$

Thus, we conclude that $L(Z^*) - L(\hat{Z}) \leq 2\Delta_n$. To achieve $\Delta_n \leq \epsilon$, we obtain the sample size bound:

$$n \geq O\left(\epsilon^{-2}\left(\mathcal{R}_n^2 + \ln\frac{1}{\delta}\right)\right), \tag{23}$$

which ensures the desired generalization bound.

A.7  METRICS FOR STSEB

STSEB compresses redundant information along the spatiotemporal dimension in SNNs, yielding more compact spatiotemporal features. To quantify the extent of spatiotemporal information compression and redundancy in the intermediate variable $Z$ within STSEB, and to facilitate a comparative analysis with traditional IB methods, we present the corresponding metrics in this section. In SNNs, LIF neurons function as core computational units, and their sparse firing activity reflects the information sparsity across layers. Typically, spike activity in higher layers becomes sparser, indicating stronger information compression. To more effectively quantify how STSEB compresses spatiotemporal information in SNNs and assess its effectiveness in removing redundant information, we introduce the following two metrics:

**Spatiotemporal Compression via FR.**  Let latent variable $Z$ correspond to $N$ LIF neurons in an SNN layer. We define the Firing Rate (FR) of $Z$ over a temporal window $T$ as:

$$\text{FR}_Z = \frac{1}{N \cdot T} \sum_{i=1}^{N} \sum_{t=1}^{T} s_i^Z(t), \tag{24}$$

where $s_i^Z(t) \in \{0, 1\}$ indicates the firing state of the $i$-th neuron at timestep $t$. Then under IB constraint, the FR of $Z$ is inversely proportional to its spatiotemporal compression degree: Lower $\text{FR}_Z \Rightarrow$ Stronger Compression.

**Spatiotemporal Redundancy via $TC(Z)$.** Let spatiotemporal variable $Z = [Z_1, \ldots, Z_d]^T \in \mathcal{M} \subset \mathbb{R}^N$ with latent representation space $\mathcal{M}$, its informational redundancy $R(Z)$ satisfies:

$$R(Z) = \alpha \cdot TC(Z) + \beta, \tag{25}$$

where $\alpha > 0$ and total correlation $TC(Z)$ is defined as:

$$TC(Z) := D_{\text{KL}} \left( \prod_{i=1}^{d} P_{Z_i} \parallel P_Z \right). \tag{26}$$

The redundancy measure $R(Z)$ exhibits strict monotonicity: $\frac{\partial R}{\partial TC} = \alpha > 0$. As $TC(Z)$ increases, the redundancy increases

## A.8 DETAILS ON ROBUSTNESS EXPERIMENTS

In the robustness evaluation section, we introduce Gaussian noise and adversarial noise during the model testing process to assess its robustness. The Gaussian noise with a mean of $\mu = 0$ and variance of $\sigma$ can be expressed mathematically as:

$$f_{\text{Gaussian}}(x) = \frac{1}{\sqrt{2\pi}\sigma} e^{-\frac{(x-\mu)^2}{2\sigma^2}}, \tag{27}$$

where $x$ is a random variable, and $\mu$ and $\sigma$ represent its mean and variance, respectively. Additionally, to better reflect the model's resistance to interference, we add two types of adversarial noise: black-box attacks and white-box attacks. For the white-box attack, we use the Fast Gradient Sign Method (FGSM), which generates adversarial noise by utilizing the gradient of the input data $x$. The expression of FGSM-based white-box attacks is:

$$\eta_{\text{white-box}} = \varepsilon \cdot \text{sign}(\nabla_x J(\theta, x, y)), \tag{28}$$

where $\varepsilon$ represents the strength of the white-box attack, and $J(\theta, x, y)$ corresponds to the model's loss function. The black-box attack, occurs when the attacker does not have access to the internal details of the model and can only observe its input-output behavior. For black-box adversarial attacks, we employ the zero-order optimization method to generate the noise. Given the input sample $x_{\text{adv}} = \text{inputs}$ and a random direction vector $\delta \sim N(0, I)$, the loss variation as the input sample moves in the direction of the vector is:

$$\Delta L = L(f(x_{\text{adv}} + \alpha\delta), \text{target}) - L(f(x_{\text{adv}}), \text{target}), \tag{29}$$

where $\alpha$ denotes the perturbation strength. Since black-box attacks cannot directly access gradient information, the loss variation along the perturbation direction is used to estimate the gradient, and update the adversarial samples as follows:

$$x_{\text{adv}} = x_{\text{adv}} + \alpha\delta \cdot \text{sign}(\Delta L). \tag{30}$$

For the generated adversarial samples, we also apply a clipping operation to ensure their similarity to the original data, which increases the effectiveness of the adversarial attack. Finally, the black-box attack samples are obtained by applying the clipping operation as follows:

$$x_{\text{adv}} = \text{clip}(x_{\text{adv}}, \text{inputs} - \epsilon, \text{inputs} + \epsilon), \tag{31}$$

where $\epsilon$ denotes the perturbation range set for the pruning operation.

## A.9 ENERGY CONSUMPTION

We theoretically analyze the impact of STSEB on SNN energy consumption, comparing it with the baseline SNN model and the SNN model with traditional IB. In SNNs, due to the pulse characteristics of their spiking neurons, the neurons are not always involved in computation. They only perform computations when the membrane potential reaches the threshold and outputs a spike. In neuromorphic chips, synaptic processing dominates the system's energy consumption and is the best indicator for evaluating the overall energy consumption of the model on the chip Furber (2016). Many studies estimate SNN energy consumption based on SOPs. Although the energy consumption generated by SOPs does not cover the actual total energy consumption on neuromorphic chips, in some systems that fully utilize sparsity (such as Loihi Davies et al. (2018)), energy consumption from other aspects

like memory access and data transmission is relatively small. The total system energy consumption is approximately proportional to SOPs, making it reasonable to use SOPs for theoretical power consumption calculations. Therefore, for ANN models, we use FLOPs to estimate their theoretical power consumption, while for SNNs, we use SOPs to evaluate their theoretical power consumption on neuromorphic chips. Specifically, FLOPs represents the number of floating-point operations per second, which increases as the number of computations and network parameters increase. The computational formula is as follows:

$$\text{FLOPs}_{\text{ANN}}(l) = H_{\text{out}} \times W_{\text{out}} \times C_{\text{in}} \times C_{\text{out}} \times K \times K, \tag{32}$$

where $H_{\text{out}} \times W_{\text{out}}$ represents the output dimension of the $l$-th layer, and $C_{\text{in}}$ and $C_{\text{out}}$ are the input and output channels. The size $K$ corresponds to the size of the convolutional kernel. For ANN, the input and output channels correspond to the learned weights. Common types of FLOP operations are multiplication-addition operations (MAC) and simple addition operations (AC). In ANNs, these operations are typically performed using the MAC method, so the theoretical power consumption can be calculated as:

$$E_{\text{ANN}} = \sum_{l} \text{FLOPs}_{\text{ANN}}(l) \times E_{\text{MAC}}. \tag{33}$$

For SNNs, we use SOPs and the energy consumption per synaptic operation $C_E = 23.6pJ$ on the Loihi chip for theoretical energy calculation. The specific calculation formula is as follows:

$$E_{SNN} = C_E \cdot SOPs = C_E \sum_{i} s_i c_i, \tag{34}$$

where $C_E$ represents the energy consumption per synaptic operation, and $SOPs$ represents the total number of synaptic operations. For each neuron $i$, $s_i$ represents the total number of spikes emitted by the neuron, and $c_i$ represents the number of synaptic connections of the neuron. Based on this theoretical estimate, we calculate the power consumption of the same structure for SNN baseline, SNN with traditional IB, and SNN with STSEB, and compare the results under two different network structures. The results are shown in Tables 4 and 5. This improvement substantiates STSEB's effectiveness in enhancing SNN energy efficiency through spatiotemporal compression.

Table 4: Energy consumption comparison on ResNet model.

| Energy consumption (pJ) | ANN | SNN | SNN+IB | SNN+STSEB |
|---|---|---|---|---|
| Layer 1 | 90,596,966.6 | 593,913,446.4 | 593,913,446.4 | 593,913,446.4 |
| Layer 2 | 1,902,536,294 | 750,257,575.8 | 712,647,321.2 | 650,722,651.3 |
| Layer 3 | 135,895,449.6 | 213,894,538.5 | 216,868,174.5 | 213,045,226.7 |
| Layer 4 | 441,660,211.2 | 779,589,003.1 | 755,348,659.5 | 696,888,632.8 |
| Total | 2,570,688,922 | 2,337,654,563.75 | 2,278,777,601.65 | 2,154,569,957.24 |

Table 5: Energy consumption comparison on VGG model.

| Energy consumption (pJ) | ANN | SNN | SNN+IB | SNN+STSEB |
|---|---|---|---|---|
| Layer 1 | 135,895,449.6 | 19,735,095.83 | 18,536,918.22 | 17,697,371.11 |
| Layer 2 | 2,174,327,194 | 408,465,901.66 | 394,430,605.37 | 365,232,047.91 |
| Layer 3 | 543,581,798.4 | 104,218,570.67 | 97,531,149.09 | 94,527,924.56 |
| Layer 4 | 135,895,449.6 | 45,297,447.11 | 44,377,739.69 | 44,038,021.71 |
| Layer 5 | 33,973,862.4 | 25,110,565.45 | 24,173,545.84 | 24,088,784.83 |
| Total | 3,023,673,754 | 602,827,580.72 | 579,049,958.21 | 545,584,150.11 |

