# OpenReview forum: "Spatiotemporal Spiking Entropic Bottleneck: Data-efficient Learning with Joint Redundancy Reduction in Spiking Neural Networks"
_ICLR.cc/2026/Conference — ICLR 2026 Conference Withdrawn Submission_

### Official Review · Reviewer_gxLU · 2025-10-21

**Soundness:** 3
**Presentation:** 2
**Contribution:** 2
**Rating:** 2
**Confidence:** 3

**Summary:**

This paper proposes STSEB (spatiotemporal spiking entropic bottleneck), which aims to utilize information bound theories to improve SNN model's performance and robustness under scarce training samples. STSEB mainly include a spike time matrix and a STM-REntB loss function based on Renyi-alpha entropy, effectively taking advantage of the spatialtemporal and binary characteristic of SNNs. Experimental results have shown the effectiveness of this method on both model generalization and robustness, and it results in low firing rate of SNNs.

**Strengths:**

1. The method proposed by this work has a good theoretical guarantee.
2. This work not only improves the model performance under scarce training samples, but also decreases SNN's firing rate and thus lowers the energy consumption.

**Weaknesses:**

See questions.

**Questions:**

1. What is the core issue this paper aim to address? Is it SNNs' performance under scarce training data? If so, is IB-based methods the only way to address this issue? Authors should clarify this, and list other paths to address few training data in related work.
2. I think the baselines in experiments are not enough. Authors have listed SIBoLS, SNIB, HOSIB, and HHO-IB in reated work, but only compares HOSIB in the experiment. The other selected baselines seems not to target at training SNNs with few training data. Also, if there are other paths to address few training data, related works should also be listed.
3. In section 5.2, I am not sure that model robustness is related to information compression.
4. In section 5.4, is the traditional IB approach implemented by the authors or sourced from another work? This should be clarified.
5. Based on the definition of $T_{nt}$ in Eq. 6, the spike time matrix (STM) $T_Z$ appears to have only one distinct value across the time dimension for each row (representing a neuron). Does it mean this matrix can actually converted into a vector?
6. Is $L_{STM-REntB}$ the final loss of this method? Which part represents the classification loss? Besides, does neurons in different layers regarded different in the final loss?

---

### Official Review · Reviewer_auot · 2025-10-24

**Soundness:** 2
**Presentation:** 2
**Contribution:** 2
**Rating:** 2
**Confidence:** 4

**Summary:**

This paper proposes the Spatiotemporal Spiking Entropic Bottleneck (STSEB) framework, which jointly compresses spatial and temporal redundancy in spiking neural networks to improve generalization and robustness under data-scarce conditions. Its core innovations include constructing a Spike Time Matrix (STM) to capture the most discriminative temporal features and designing a Rényi α-entropy estimator to quantify and suppress joint spatiotemporal redundancy. The authors further provide theoretical analyses of generalization bounds and sample complexity. Experimental results on datasets such as DVS-Gesture and CIFAR-10/100, including few-shot training and evaluations under noise and adversarial attacks, demonstrate superior data efficiency, enhanced robustness, and reduced energy consumption.

**Strengths:**

1. By jointly modeling and compressing spatial and temporal redundancy through STM and Rényi entropy, the proposed method differs from prior IB-based SNN approaches that focus solely on spatial compression.

2. The authors provide rigorous proofs of generalization bounds and sample complexity, offering strong theoretical support, and conduct extensive comparative and ablation experiments.

**Weaknesses:**

1. STSEB does not exhibit a clear advantage in few-shot scenarios. In other words, Figure 4 only demonstrates that STSEB performs better overall, but fails to reveal any remarkable superiority of STSEB specifically under limited data conditions.

2. To the best of my knowledge, TCJA is a spatiotemporal attention module rather than a training optimization method. Therefore, the corresponding description near line 372 should be revised for accuracy.

3. The temporal encoding adopted in this study naturally reduces redundancy, spike counts, and power consumption. Thus, the more critical scientific question concerns why fewer spikes can lead to better performance. It is recommended to deepen the analysis of neuronal dynamics when integrating STSEB, especially how such dynamics evolve and are optimized during training.

**Questions:**

1. The authors may have chosen TCJA as the primary comparison due to its 99.0% accuracy on the DVS-Gesture dataset. However, why not include comparisons with Spike-driven Transformer V1/V2, which achieve even higher accuracies on the same dataset? Such comparisons would further validate the generalizability of the proposed method within Transformer-based architectures.

2. The evaluation is limited to a single, relatively simple event-based classification dataset and two static image classification datasets. This experimental scope may not be sufficient to demonstrate the true effectiveness of STSEB. More importantly, this raises concerns that STSEB might only achieve optimal performance due to fine-tuned parameter configurations on these simpler tasks. It is strongly recommended to include experiments on medium-scale datasets such as ImageNet, CIFAR10-DVS, or N-Caltech101 to verify whether STSEB remains effective at a more realistic scale.

---

### Official Review · Reviewer_RWAn · 2025-10-29

**Soundness:** 4
**Presentation:** 2
**Contribution:** 3
**Rating:** 6
**Confidence:** 5

**Summary:**

This paper presents spatiotemporal spiking entropic bottleneck to quantify the redundancy of the SNNs. By utilizing this framework, the researches show that we can get a more compressed yet representative and robust SNNs. Overall, this is a novel research for lightweight SNNs and provided a new perspective.

**Strengths:**

1. Innovative Spatiotemporal Joint Compression Mechanism: The key idea is the use of a Spike Time Matrix (STM) that records only the first spike time of each neuron, preserving essential temporal features while discarding repeated spikes. A Rényi’s α-entropy estimator is then applied to quantify spatiotemporal information and guide compression.
2. Strong Generalization, Robustness, and Energy Efficiency: STSEB demonstrates significant improvements in generalization, robustness, and energy consumption across multiple benchmarks.

**Weaknesses:**

1. The logic of the abstract is questionable: while emphasizing the shortcomings of SNNs, the authors focus on "data scarce" but later shift the focus to "information bottleneck" and "redundant metrics." Although the authors' experiments show some robustness to few-shot learning, the overall structure of the article is unclear. The authors should further clarify the relationship between "data scarce" and "representation redundancy" in the related works or introduction.
2. Limited experimental models: The authors only designed STSEB for TCJA, neglecting the importance of the framework's transferability and extensibility. They should apply this framework to other mainstream SNNs and evaluate their compression benefits. To my knowledge, similar IB theory can also be applied to the Transformer architecture; the authors should further explore its applicability to the Spiking Transformer architecture.
3. Tables 4 and 5 in the appendix suggest using higher-order units of energy consumption for easier reading.

**Questions:**

1. Why does the spatiotemporal redundancy problem of SNN pulse coding only occur when data is scarce? Will other coding problems arise in cases of long-tailed distribution, uneven distribution, or other situations?
2. Eq. 5 shows that this framework mainly targets the temporal characteristics of SNN neurons (time-biased), but I did not see any formulas regarding spatial information compression. Could the author please clarify whether there is any compression of spatial features? If I have misunderstood anything, please point it out.
3. Why choose Renyi‘s Entropy to construct the framework, rather than other information metrics (e.g.: Matrix-Based Entropy[1])

[1].Layer by Layer: Uncovering Hidden Representations in Language Models, Oscar Skean, et al., ICML 2025

---

### Official Review · Reviewer_L7ce · 2025-10-31

**Soundness:** 4
**Presentation:** 4
**Contribution:** 4
**Rating:** 8
**Confidence:** 5

**Summary:**

This paper proposes a new Spatiotemporal Spiking Entropic Bottleneck (STSEB) framework to address the poor generalization and limited robustness of Spiking Neural Networks (SNNs) under data-scarce conditions. The core idea of this framework is to construct a Spike Time Matrix and integrate a Rényi’s α-entropy estimator to achieve joint compression in both spatial and temporal dimensions of SNNs, thereby reducing redundancy and enhancing the model’s data utilization efficiency. Experiments on multiple datasets demonstrate that the proposed method achieves high data efficiency, strong generalization, and improved robustness while reducing the model’s energy consumption.

**Strengths:**

1. This paper innovatively proposes a new framework called STSEB, which ingeniously achieves joint compression of SNNs in both spatial and temporal dimensions through the construction of a Spike Time Matrix. This approach significantly enhances the generalization and robustness of SNNs in small-sample scenarios, enabling more effective information compression.
2. The paper is well written and easy to follow. The proposed method is grounded in solid mathematical foundations, supported by rigorous and comprehensive theoretical derivations and visualization analyses.
3. Experiments cover both static and neuromorphic datasets, demonstrating that STSEB outperforms existing approaches in terms of generalization, robustness, and data efficiency while reducing the energy consumption of SNNs.
4. STSEB aligns well with the spiking and event-driven characteristics of SNNs, making it valuable both theoretically and practically.

**Weaknesses:**

1. Through experimental comparisons, STSEB demonstrates superior performance over the traditional IB. However, the specific experimental setup details of the compared traditional IB are not clearly provided. Supplementing this information would strengthen the claim of STSEB’s superiority.
2. The hyperparameters in STSEB affect the compression performance of spatiotemporal information, yet their specific settings are not detailed in the appendix, which may hinder the reproducibility of the results.
3. The extensive experiments conducted on both static and neuromorphic datasets are valuable and sufficient. Nevertheless, future validation on other tasks (such as optical flow estimation, which is more suitable for SNNs) would further reinforce the claims regarding STSEB’s generalization and robustness.

**Questions:**

See weaknesses 1–2.

---

### Note · Authors · 2025-11-28

I have read and agree with the venue's withdrawal policy on behalf of myself and my co-authors.